# Short- and Long-Term Outcomes After Emergency Groin Hernia Surgery: A Nationwide Population-Based Study from the Swedish Hernia Register

**DOI:** 10.3390/jcm14072397

**Published:** 2025-03-31

**Authors:** Maria Melkemichel, Henrik Holmberg, Ursula Dahlstrand, Hanna de la Croix

**Affiliations:** 1Department of Clinical Science and Education, Södersjukhuset, Karolinska Institutet, 11883 Stockholm, Sweden; maria.melkemichel@ki.se; 2Department of Breast, Endocrine and Sarcoma Tumors, Karolinska University Hospital, 17164 Stockholm, Sweden; 3Department of Epidemiology and Global Health, Umeå University, 90187 Umeå, Sweden; henrik.holmberg@umu.se; 4Department of Clinical Science, Intervention and Technology, Karolinska Institutet, 11883 Stockholm, Sweden; ursula.dahlstrand@ki.se; 5Department of Surgery, Enköping Hospital, 74538 Enköping, Sweden; 6Department of Surgery, Institute of Clinical Sciences, Sahlgrenska Academy, University of Gothenburg, 40530 Gothenburg, Sweden; 7Department of Surgery, Sahlgrenska University Hospital, Region Västra Götaland, 41650 Gothenburg, Sweden

**Keywords:** emergency groin hernia repair, mortality, bowel resection, chronic pain, recurrence, postoperative complications

## Abstract

**Background/Objectives**: Emergency groin hernia repairs have consistently presented a higher risk of mortality and morbidity. This study aimed to compare both short- and long-term outcomes associated with emergency groin hernia surgery. **Methods**: A nationwide, population-based cohort study was conducted using prospective collected data from the Swedish Hernia Register combined with a questionnaire assessing patient-reported chronic pain. All patients who underwent a groin hernia repair between 2012 and 2018 were eligible for inclusion. Primary outcomes included 30-day mortality, chronic pain 1-year post-surgery, 30-day postoperative complication, and bowel resection and reoperation for recurrence for emergency versus elective repairs. Risk factors for these outcomes in emergency repair were investigated. **Results**: A total of 94,349 repairs were analyzed, with 5401 (5%) emergency repairs. Emergency repairs involved older patients (median age 74 vs. 65), more women (25% vs. 9%), more ASA grade III (38% vs. 12%), more femoral hernias (19% vs. 3%) and smaller defects (24% vs. 17%) compared to elective repairs. Multivariable analysis revealed increased rates and significant risks for 30-day mortality (2.7%, OR 11.61), chronic pain (20.6%, OR 1.30), 30-day postoperative complications (21.9%, OR 2.12) and bowel resection (7.8%, OR 408) compared to elective repairs. No significant difference was observed for reoperation for recurrence. Key risk factors for the outcomes following emergency repairs were higher age, higher ASA grade and femoral hernias. **Conclusions**: Emergency hernia surgery continues to pose a high risk of mortality and morbidity. Elective repair should be considered in frail patients and those with potential femoral hernias.

## 1. Introduction

Emergency groin hernia repair carries a significant risk of both morbidity and mortality [1,2,3]. However, the urgent nature of these procedures makes them challenging to study effectively. In Western countries, only a small percentage of groin hernia repairs are conducted as emergency surgery, while the rates are considerably higher in sub-Saharan Africa [4,5]. As the only treatment available for strangulated hernias, delays in surgical intervention are linked to increased morbidity and mortality [6]. Consequently, the global prevalence of emergency hernias underscores the importance of investigating this issue to enhance patient outcomes.

Current international guidelines for groin hernia management highlight the scarcity of evidence surrounding emergency repairs and call for future studies to focus on factors contributing to incarceration and strangulation [7,8]. While bowel resection can often be avoided if the strangulation has lasted shorter than six hours, the need for resection can be used as an indication of irreversible strangulation ischemia. Previous research has indicated that female gender and femoral hernias elevate the risk of requiring emergency surgery with strangulation [9,10]. Moreover, mortality following groin hernia repair is associated with increasing age, strangulation, comorbidities and female gender [11,12]. Studies from both the Danish Database and the national Swedish Hernia Register have included emergency repair data from before the millennium; however, these are from an era that is different in terms of both surgical methods and perioperative care compared to now [2,3].

Additionally, few studies, if any, have examined patient-reported outcome measures concerning chronic pain following emergency hernia repair. In the Swedish Hernia Register, approximately 5% of all groin hernia repairs are performed as emergency operations, with the rate for women reaching around 12% [13]. The register prospectively registers all operations, and between 2012 and 2018, all patients were invited to complete a questionnaire assessing chronic postoperative pain.

This study aimed to evaluate and compare emergency and elective groin hernia repairs regarding 30-day mortality; patient-reported chronic pain 1-year post-surgery; 30-day postoperative complication; bowel resection; and reoperation for recurrence. A secondary outcome involved assessment of risk factors associated with these outcomes following emergency groin hernia repair. By integrating contemporary national data from the Swedish Hernia Register with patient-reported outcome measures (PROMs), this study seeks to provide a comprehensive assessment of outcomes related to emergency groin hernia repairs in current clinical practice.

## 2. Material and Methods

### 2.1. Study Design

This nationwide, population-based register cohort-study utilized prospectively collected data from the Swedish Hernia Register (SHR) combined with patient-reported outcome measures (PROMs) between 2012 and 2018 to evaluate chronic pain 1 year after surgery and reoperation for recurrence as well as 30-day mortality, perioperative bowel resection and 30-day postoperative complication for emergency and elective groin hernia repairs. The study adhered to the STROBE guidelines [14].

### 2.2. Swedish Hernia Register

Since its inception in 1992, the SHR has prospectively documented more than 400,000 groin hernia repairs [13]. Each year, an external reviewer from the SHR verifies the data [15]. National data completeness was achieved in 2004, and validations of the register indicate a data accuracy of 98% [15]. Surgeons record detailed information regarding patient demographics, surgical indications, hernia anatomy and repair methods at the time of the surgery, prospectively.

The register is linked to the Migration and Cause of Death Registry in Sweden, allowing for precise follow-up on reoperations, emigrations or deaths, facilitated by each patient’s unique personal identity number. Between 2012 and 2018, chronic pain was evaluated using a patient-reported outcome measure (PROM) questionnaire incorporated into the registry. This questionnaire was distributed 1-year post-surgery to all patients treated at participating units within the SHR. It includes a brief pain assessment derived from the validated Inguinal Pain Questionnaire (IPQ) and has been shown to provide comparable information to the original IPQ [16].

### 2.3. Study Population

All groin hernia repairs performed between 1 January 2012 and 31 October 2018, aged 15 and above, were eligible for inclusion (Figure 1). Repairs with an unspecific hernia anatomy and bilateral repairs were excluded (Figure 1). Included hernia repairs consisted of emergency and elective groin hernias performed with different surgical methods of repair: open anterior mesh repair (according to the Lichtenstein procedure), laparo-endoscopic repair (TEP (totally extraperitoneal) and TAPP (transabdominal preperitoneal plasty)), open posterior mesh repair (TIPP (transinguinal preperitoneal repair), Stoppa and Nuhys), combined anterior/posterior mesh repair (bilayer mesh and predominantly mesh plugs) and non-mesh suture repair (predominately Shouldice and Bassini repair) (Table 1). Emergency versus elective repair was registered as a dichotomous variable (emergency/elective) in the SHR, where emergency repair was defined as an unplanned repair within 24 h due to acute symptoms of hernia complication. Comorbidity was recorded according to the ASA physical fitness grade, which refers to the Physical status classification system adopted in 1963 by the American Society of Anesthesiologists. Grades I–II refer to persons who are heathy or suffering from a mild systemic disease, and grades III–IV refer to persons who are suffering from severe systemic diseases, some of which are life threatening.

The study period was based on when the patient-reported outcome measure (PROM) questionnaire was incorporated into the registry and to represent a current contemporary practice of repairing emergency groin hernias with up-to-date outcome measurements. Clinically important variables are presented in Table 1.

### 2.4. PROM Pain Questionnaire

Chronic pain was defined as persisting significant pain 1 year after surgery. Item 2 from the IPQ was sent out to all operated patients with instructions to “rate your worst pain in the operated groin during the previous week” using a 7-point scale. Pain was graded from 1 to 7: level 1—no pain; level 2—pain present, but easily ignored; level 3—pain present, cannot be ignored but does not interfere with everyday activities; level 4—pain present, cannot be ignored and interferes with concentration on everyday activities; level 5—pain present, interferes with most activities; level 6—pain present, necessitating bed rest; and level 7—pain present, prompt medical advice sought. Scores on levels 1–3 were in this study defined as No pain, whereas scores on levels 4–7 were defined as Pain, indicating significant postoperative chronic pain in accordance with other studies [17]. Non-responders were sent a reminder within 30 days.

### 2.5. Study Objectives

The primary outcomes included short- and long-term postoperative complications after emergency versus elective surgery.

Thirty-day mortality (I): Registered as a dichotomous variable (yes/no) in the SHR based on information from the hospital medical charts and link to the cause of death registry.

Patient-reported chronic pain 1-year post-surgery (II): Chronic pain was defined as persisting significant pain 1 year after surgery, assessed via a patient-reported outcome measure (PROM) questionnaire incorporated into the registry. Only those who responded to the PROM pain questionnaire were included in the analysis for chronic pain (Table 1).

Thirty-day postoperative complications (III): Postoperative complications were registered in the SHR prospectively, 30 days after surgery, as a dichotomous variable (yes/no) with information from the hospital medical charts. Included complications were seroma, hematoma, surgical site infections, urinary retention, bleeding, and cardio-vascular and pulmonary conditions.

Bowel resection (IV): Registered perioperatively as a dichotomous variable (yes/no) in the SHR.

Reoperation for recurrence (V): Reoperation for recurrence was characterized as a subsequent hernia repair in the same groin where the prior index repair was conducted and documented in the SHR database. Only a single reoperation for recurrence in each groin within each patient was considered. Reoperation for recurrence was investigated with a follow-up time extending to 6 November 2020.

The secondary outcome involved associated independent risk factors of above outcomes (I–V) for the population of emergency groin hernia repairs (Table 3).

### 2.6. Statistical Analysis

Descriptive statistics are presented in Table 1. Categorical variables are described with numbers and percentages. Continuous variables are presented with medians and range. Multivariable logistic regression analyses were performed to estimate odds ratios (OR) for the risk of 30-day mortality, chronic pain, 30-day postoperative complications and bowel resection. Cox proportional analysis was undertaken to estimate the hazard ratio (HR) for the risk of reoperation for recurrence (Table 2 and Table 3). Adjustments were made for age, ASA fitness grade, hernia anatomy, primary/recurrent hernia, surgical method of repair, bowel resection and postoperative complication (Table 2 and Table 3). Time from the index repair until reoperation for recurrence in years was used to calculate the cumulative reoperation rate for recurrence, also described in a reciprocal Kaplan–Meier curve (Figure 2). The 95% confidence intervals (CI) were calculated and statistical significance was set to *p* < 0.05. All statistical analyses were performed using R Core Team 2022 (Version 4.4.1).

## 3. Results

### 3.1. Baseline Characteristics

Between 1 January 2012 and 31 October 2018, a total of 94,349 groin hernia repairs were recorded in the SHR. Among these, 5137 (5.4%) were performed as emergency repairs (Table 1, Figure 1). 

Their baseline characteristics are detailed in Table 1. The emergency repair group included a higher proportion of elderly patients, with a median age of 74.3 years compared to 64.6 years in the elective repair group. Additionally, there were higher proportions of women (24.9% vs. 8.9%), recurrent hernias (14.9% vs. 8.4%) and patients classified as ASA fitness grades III–IV (37.9% vs. 12.0%) among those undergoing emergency repairs compared to elective repairs. The emergency group also had a higher occurrence of femoral hernias (18.6% vs. 2.8%) and small defects < 1.5 cm (23.9% vs. 17.0%) compared to those undergoing elective repairs (Table 1). In terms of surgical methods of repair, open posterior mesh and suture repairs were more often used in emergency repairs than in elective repairs (Table 1).

Emergency repairs were associated with a higher 30-day mortality rate (2.7% vs. 0.1%) compared to elective repairs (Table 1). Correspondingly, 72.3% (n = 141) of all repairs with 30-day mortality (n = 195) consisted of emergency repairs. The adjusted multivariable analysis revealed significant eleven-fold increased odds for 30-day mortality for emergency repairs, with an OR of 11.61 (95% CI 8.09–16.67) compared to elective repairs (Table 2).

### 3.2. Chronic Pain

The PROM pain questionnaire was completed by 60,618 patients (9343 for emergency repairs and 51,275 for elective repairs), resulting in an overall response rate of 64.2% (Table 1). Among them, 9343 (15.4%) patients reported chronic pain 1-year post-surgery. Out of those who reported pain, 500 had undergone an emergency repair. There was a 20.6% reported chronic pain rate and a response rate of 47.3% for emergency repairs compared to 15.2% reported chronic pain and a response rate of 65.2% for elective repairs (Table 1). Multivariable logistic regression analysis revealed that emergency repairs had significantly higher odds of chronic pain compared to elective repairs with an OR of 1.30 and 95% CI 1.17–1.45 (Table 2).

### 3.3. 30-Day Postoperative Complication

Emergency repairs had a higher rate of 30-day postoperative complication than elective repairs (21.9% vs. 8.8%) (Table 1). In the analysis, a significantly increased two-fold OR at 2.12 (95% CI 1.97–2.30) was noticed (Table 2).

### 3.4. Bowel Resection

Almost all bowel resections in the cohort were performed within the emergency repair population with a total rate of 7.8% of the emergency repairs compared to 0% for elective repairs (Table 1). The odds of bowel resection for emergency repairs compared to elective repairs was estimated to an OR at 408 (95% CI 179.92–926.40) (Table 2).

### 3.5. Reoperation for Recurrence

The reoperation rate for recurrence was similar in both groups: 2.8% vs. 2.5% (Table 1), with a non-significantly increased hazard ratio at 1.14 (95% CI 0.95–1.38) for emergency repairs (Table 2). An unadjusted curve of cumulative reoperation for recurrence for elective versus emergency repairs is illustrated in Figure 2.

### 3.6. Risk Factors for Analysed Outcomes Among Emergency Repairs

Risk factors for the various outcomes associated with emergency repairs are presented in Table 3. Factors linked to an increased risk of 30-day mortality included age above the median (>65 years), ASA fitness grades III–V, non-mesh suture repair, the presence of 30-day postoperative complication and the need for bowel resection. The risk of chronic pain was associated with age over 65 years and ASA fitness grades III–V. Additionally, ASA fitness grades III–V, femoral hernias, hernia defects > 3 cm, non-mesh suture repairs and bowel resection were all identified as risk factors for 30-day postoperative complication. The likelihood of requiring bowel resection during emergency repairs was associated with age over 65 years, ASA fitness grades III–V, femoral hernias, open anterior repairs, combined mesh repair, suture repair and the presence of 30-day postoperative complication. The risk of reoperation for recurrence was associated to prior recurrent hernia repairs and the presence of 30-day postoperative complication.

## 4. Discussion

Based on a nationwide modern cohort of nearly 95,000 groin hernia repairs, this study reveals that the 30-day mortality risk is still significantly increased, by a factor of 11, following emergency surgery compared to elective repair. The risk of bowel resection in emergency surgery is greatly heightened, multiplied by hundreds, while the risk of postoperative complications is doubled and the likelihood of experiencing chronic pain is significantly elevated compared to elective repair. Furthermore, the risk of bowel resection among those undergoing emergency repairs is increased for elderly patients, those with comorbidities and individuals with femoral hernias. However, the risk of reoperation due to recurrence does not appear to be influenced by the emergency setting compared to the elective one in this contemporary analysis.

The 30-day mortality risk for emergency repairs compared to elective repairs in this modern study period aligns with a study from 2007, showing that patients undergoing elective hernia repair experience a lower mortality rate compared to an age- and sex-matched population [3]. Despite emergency repairs accounting for 72% of all deaths within 30 days, the emergency surgery rate in this cohort was lower than in previous studies [2,3]. Notably, only 2.5% of the emergently operated patients died within 30 days of surgery, which is a lower number than the 3% reported in a previous study and in a Danish study who found a mortality rate of 6% within 30 days [2,3]. The observed decrease in mortality rates in this Swedish cohort could be attributed to improvements in perioperative care and a heightened awareness of emergency hernias in women over the past two decades. This hypothesis is supported by a Danish study that revealed a decreasing mortality trend over time, with a corresponding mortality rate of 7% in 2010 [1]. However, the discrepancy in mortality rates between this study and previous ones could also be due to differences in data collection methods. The Swedish Hernia Register enrolls patients at the time of surgery, which might mean that patients with strangulated hernias who died before arriving at the operating theatre are not included in the register. Patients in the emergency repair cohort in this study were older (by an average of 10 years) and had more comorbidities compared to those with elective repairs. The observed elevated mortality rate among older patients is consistent with a study showing a mortality rate of 1.2–6% among patients over the age of 75 years [18].

Chronic pain after emergency hernia surgery has previously been scantly studied. This study indicated a higher prevalence of chronic pain in patients who had emergency surgery compared to those who had elective procedures, which is in contrast with the findings of Dahlstrand et al., where femoral emergency hernia repairs were associated with a decreased risk of pain [19]. It is important to note that Dahlstrand et al. focused exclusively on femoral hernias. In this larger-scale cohort, however, patients with femoral emergency hernias did not report less pain than those with inguinal hernias. Further studies with data on patient-reported outcomes following emergency hernia surgery would be beneficial.

The rate and risk of bowel resection following emergency repair, compared to elective repairs, were increased in this cohort. The present study reported a bowel resection rate of 7.8%, in contrast to a Danish study that revealed a halved rate of 3.9% [2]. The reason for this discrepancy in rates is unclear and may relate to differences in the definition of emergency groin hernia repair. In the Swedish Hernia Register, emergency surgery is defined as an urgent hernia repair performed within 24 h of hospital admission. Conversely, in the Danish Hernia Database, the surgeon determines whether the procedure qualifies as an emergency when documenting it in the register. As a result, a hernia without signs of strangulation may still be operated on after a few days due to associated pain. According to international guidelines for the management of groin hernias, risk factors for strangulation include female gender, femoral hernia and a history of previous hospitalization due to the groin hernia [7]. Even though the proportion of women that underwent emergency surgery (24.9%) was much higher than women who underwent elective repairs (8.9%), female gender was not an associated factor for the risk of bowel resection following emergency surgery (Table 3). In this study, femoral hernia, older age and increased comorbidities were associated with a higher risk of bowel resection. Interestingly, neither hernia defect size nor the history of recurrent repair showed any impact on the risk of bowel resection. 

The use of mesh in emergency hernia repair has been a topic of ongoing debate, largely due to concerns that contaminated areas could lead to mesh infections. The guidelines conclude that there are insufficient data to definitively support either approach in emergency hernia surgery [7,8]. In non-contaminated settings, traditional methods of repair are recommended [8] and the use of prosthetic mesh in non-contaminated emergency settings is generally considered safe [20]. A small retrospective study by Kumar et al. demonstrated no significant difference in wound infection rates, recurrences or reoperations between mesh and non-mesh repair following emergency groin hernia procedures [21]. In contrast, Shi et al. reported a substantial difference in recurrence rates with non-mesh suture repairs, although the postoperative complication rates were similar between the two techniques [22]. Additionally, a large Cochrane review found no significant differences in outcomes due to a lack of sufficient evidence [23]. Echoing the findings of Shi et al., a study based on the Danish Hernia Database indicated that non-mesh techniques were associated with higher reoperation rates [2]. In this present study, non-mesh suture repairs were relatively rare, used in only 5% of emergency hernia operations. Among the emergency cohort, laparo-endoscopic techniques were associated with a lower rate of bowel resection, whereas non-mesh suture repairs were linked to increased mortality and bowel resection rates. This disparity is likely to be attributed to the selection of surgical method according to the severity of the case, where less complicated and less contaminated emergency hernias were treated with laparo-endoscopic techniques, while non-mesh suture repairs were favored in more complex and contaminated cases. However, the widespread use of mesh techniques without a corresponding increase in complications or mortality rates supports that mesh repair may be considered acceptable even in emergency hernias with contamination. 

Although emergency repairs in this study were associated with elevated risks of mortality, chronic pain, postoperative complications and bowel resections compared to elective repairs, they did not demonstrate an increased risk of reoperation for recurrence. It is challenging to compare these modern data with earlier results from studies utilizing the Swedish Hernia Register of reoperations for recurrence, as most of these studies exclude emergency repairs from the analysis due to their complexity and heterogeneity. Notably, being a recurrent repair and experiencing a postoperative complication significantly increased the risk of reoperation for recurrence following an emergency repair. 

Previous studies have consistently shown that emergency hernia repairs are associated with poorer outcomes, suggesting that we should actively work to lower the number of hernia emergencies. However, the question of whether elective hernia repair can effectively reduce the risk of emergency situations remains debatable. It is established that watchful waiting for minimally symptomatic inguinal hernias is a safe approach in men, as evidenced by the low incidence of emergency hernias developing during the observation period [24,25,26]. Furthermore, a study by Dahlstrand et al. found that a significant proportion of patients presenting with emergency femoral hernias experienced their first symptoms at the time of incarceration, highlighting the unexpected nature of these emergencies [27]. That being said, all diagnosed femoral hernias should be repaired electively, and for older and frail patients with suspicion of femoral hernia, imaging should be used to rule that out before deciding on a conservative approach. The findings of previous studies, in conjunction with the results from this cohort, suggest that the risks and benefits of elective hernia repair should be discussed with older patients and those with comorbid conditions, even if their inguinal hernia is asymptomatic. Notably, for patients where general anesthesia is not feasible, hernia repair using local anesthesia has been shown to yield good outcomes, offering a viable alternative [28].

To date, this is the largest observational register cohort study combining register data with patient-reported outcome measures assessing chronic pain after emergency groin hernia repair. A strength of the study is the cohort with near-complete national coverage of groin hernia repairs performed in Sweden. The study presents a real-life setting, and the significant number of unselected groin hernia repairs conducted by general surgeons with varying levels of expertise reduces the potential single-surgeon bias and enhances the external validity of the findings. This study’s investigation, dependent on a sample size of approximately 5000 emergency repairs, where almost half responded to a PROM questionnaire assessing chronic pain, would be challenging to accomplish without utilizing a comprehensive national population-based database cohort. Furthermore, the prospective collection of data within the registry strengthens the association between the variables and outcomes. Additionally, the study benefits from a clear definition of chronic pain, employing a question derived from the highly validated IPQ. This short-form pain questionnaire has been shown to be more user-friendly in clinical practice than the full version of the IPQ, while still providing comparable information.

However, there are some limitations to the study. As with all observational cohort studies, while correlations can be identified, caution needs to be exercised when discussing causality. One limitation of the present study is the definition of emergency surgery, which is defined as surgery within 24 h due to hernia complication. This definition varies among different studies, making it challenging to compare emergency repair rates effectively. Additionally, the low response rate to the chronic pain questionnaire in the emergency repair group represents another significant limitation. A previous study from the SHR, examining the same pain PROM through sensitivity testing, indicated that the proportion of chronic pain among non-responders is lower than that among responders, implying that non-responders may experience less pain than those who completed the questionnaire [17]. Consequently, the overall rate of chronic pain identified in the current study may be overestimated, and it remains uncertain whether this overestimation is consistent within the emergency repair group. The lack of response in this group could rather be influenced by factors such as advanced age, increased comorbidity and the presence of increased mortality in the emergency repair group compared to the elective group. Another limitation of the study is the lack of data on preoperative pain, which presents a potential confounding variable that could not be controlled for.

## 5. Conclusions

Emergency groin hernia surgery carries an associated increased risk of mortality, bowel resection, chronic pain and postoperative complications compared to elective repairs. The increased risks and the consequences thereof are particularly dire in elderly and comorbid patients. When determining whether an individual patient should undergo elective surgery, it is essential to consider these results and assess the risk for future acute incarceration. If general anesthesia is deemed problematic, local anesthesia may be a viable alternative. Both women and men with a potential femoral hernia should undergo elective surgery and, if they present with an emergent onset, immediate intervention is warranted to minimize the need for bowel resection. When bowel resection is needed during an emergency hernia repair, the risk of mortality and complication nearly doubles.

## Figures and Tables

**Figure 1 jcm-14-02397-f001:**
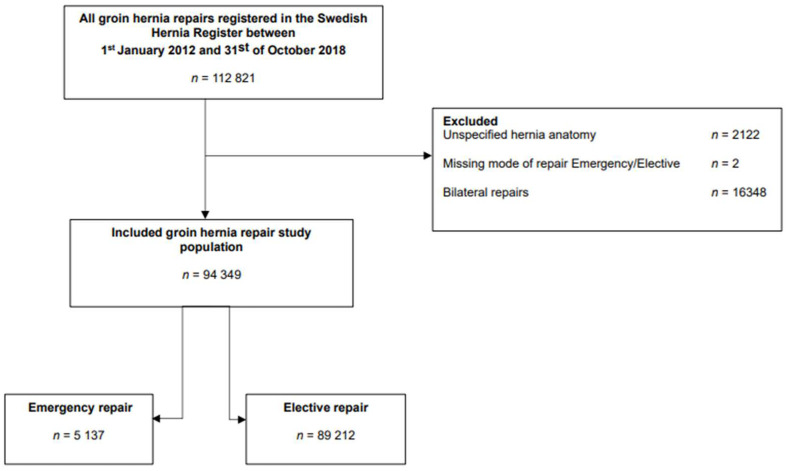
Flowchart of the included hernia repair study population.

**Figure 2 jcm-14-02397-f002:**
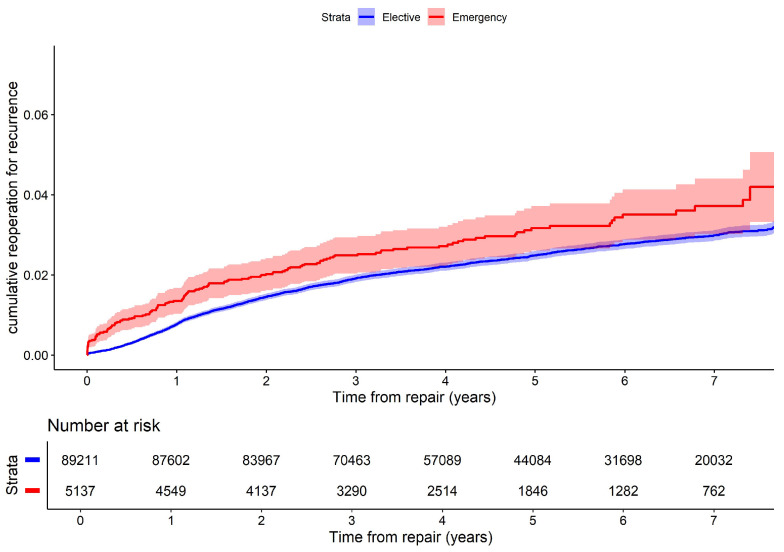
Reciprocal Kaplan–Meier curve of cumulative reoperation rate for recurrence for emergency versus elective repairs.

**Table 1 jcm-14-02397-t001:** Characteristics of emergency versus elective hernia repair study population.

	Emergency	Elective	Overall	*p* Value ^v^
Characteristics	N = 5137 (5.4%)	N = 89,212 (94.6%)	N = 94,349	
Age (years)				
Median [Min, Max]	74.3 [16.1, 102]	64.6 [15.0, 102]	65.0 [15.0, 102]	<0.001
Gender				
Man	3860 (75.1%)	81,242 (91.1%)	85,102 (90.2%)	<0.001
Woman	1277 (24.9%)	7970 (8.9%)	9247 (9.8%)	
ASA fitness grade ^i^				
I–II	3190 (62.1%)	78,514 (88.0%)	81,704 (86.6%)	<0.001
III–V	1947 (37.9%)	10,698 (12.0%)	12,645 (13.4%)	
Hernia anatomy				
Femoral	957 (18.6%)	2495 (2.8%)	3452 (3.7%)	<0.001
Combined	337 (6.6%)	7035 (7.9%)	7372 (7.8%)	
Indirect	2388 (46.5%)	51,504 (57.7%)	53,892 (57.1%)	
Direct	1455 (28.3%)	28,178 (31.6%)	29,633 (31.4%)	
Hernia defect size (cm)				
<1.5	1226 (23.9%)	15,169 (17.0%)	16,395 (17.4%)	<0.001
1.53	1536 (29.9%)	26,426 (29.6%)	27,962 (29.6%)	
>3	2370 (46.1%)	47,584 (53.3%)	49,954 (52.9%)	
Missing	5 (0.1%)	33 (0.0%)	38 (0.0%)	
Surgical method of repair				
Open anterior(a) mesh	3444 (67.0%)	66,487 (74.5%)	69,931 (74.1%)	<0.001
Laparo-endoscopic mesh	394 (7.7%)	15,398 (17.3%)	15,792 (16.7%)	
Open posterior(p) mesh	759 (14.8%)	2805 (3.1%)	3564 (3.8%)	
Combined a/p mesh	137 (2.7%)	3393 (3.8%)	3530 (3.7%)	
Non-mesh suture	294 (5.7%)	614 (0.7%)	908 (1.0%)	
Missing	109 (2.1%)	515 (0.6%)	624 (0.7%)	
Primary hernia repair				
Yes	4370 (85.1%)	81,691 (91.6%)	86,061 (91.2%)	<0.001
No (recurrent)	767 (14.9%)	7521 (8.4%)	8288 (8.8%)	
30-day postoperative complication				
Yes	1126 (21.9%)	7835 (8.8%)	8961 (9.5%)	<0.001
No	4011 (78.1%)	81,377 (91.2%)	85,388 (90.5%)	
Reoperation for recurrence ^ii^				
Yes	142 (2.8%)	2214 (2.5%)	2356 (2.5%)	0.224
No	4995 (97.2%)	86,998 (97.5%)	91,993 (97.5%)	
Bowel resection				
Yes	401 (7.8%)	7 (0.0%)	408 (0.4%)	<0.001
No	4434 (86.3%)	43,285 (48.5%)	47,719 (50.6%)	
Missing	302 (5.9%)	45,920 (51.5%)	46,222 (49.0%)	
Chronic pain 1 year ^iii,iv^				
Yes	500 (20.6%)	8843 (15.2%)	9343 (15.4%)	<0.001
No	1928 (79.4%)	49,347 (84.8%)	51,275 (84.6%)	
Non-responders	2709 (52.7%)	31,022 (34.8%)	33,731 (35.8%)	
30-day mortality				
Yes	141 (2.7%)	54 (0.1%)	195 (0.2%)	<0.001
No	4996 (97.3%)	89,158 (99.9%)	94,154 (99.8%)	

Numerical characteristics are reported as median value with range, minimum–maximum. Nominal and ordinal characteristics are reported as frequency. Percentages are presented in parentheses unless indicated otherwise. ^i^ The ASA physical fitness grade refers to the Physical status classification system adopted in 1963 by the American Society of Anesthesiologists. Grades I–II refer to persons who are healthy or suffering from a mild systemic disease, and grades III–IV refer to persons who are suffering from severe systemic diseases, some of which are life threatening. ^ii^ Until 6 November 2020. ^iii^ Individuals reporting an Inguinal Pain Questionnaire (IPQ) item 2 score greater than 3 are classified as experiencing chronic postoperative pain. ^iv^ Yes + no = responders, non-responders are missing values. ^v^
*p* values; median age is tested using Mann–Whitney U test, and categorical variables are tested using Chi2 test.

**Table 2 jcm-14-02397-t002:** Risk of 30-day mortality, chronic pain, 30-day postoperative complication, bowel resection and reoperation for recurrence—multivariable regression analyses.

Outcomes	Elective	Emergency (95% CI)	*p* Value
30-day mortality ^ (n = 93,686)	1.0 (ref)	OR 11.61 (8.09, 16.67)	**<0.001**
Chronic pain ^ (n = 60,242)	1.0	OR 1.30 (1.17, 1.45)	**<0.001**
30-day postoperative complication ^^i^ (n = 93,686)	1.0	OR 2.12 (1.97, 2.30)	**<0.001**
Bowel resection ^^i^ (n = 93,686)	1.0	OR 408.27 (179.92, 926.40)	**<0.001**
Reoperation for recurrence *^i^ (n = 93,686)	1.0	HR 1.14 (0.95, 1.38)	0.158

^ Logistic regression model with OR (odds ratio). * Cox proportional hazard regression model with HR (hazard ratio), follow-up until 6 November 2020. Age under/above the total median, gender, ASA fitness grade, hernia anatomy, primary or recurrent hernia, surgical method of repair, bowel resection and postoperative complications were adjusted for. ^i^ Adjustment for same variables except postoperative complications or bowel resection, respectively. Significant *p* (*p*-values) are in bold.

**Table 3 jcm-14-02397-t003:** Multivariable regression analyses of risk factors for outcomes following emergency repairs.

Factors	30-Day Mortality ^i^	Chronic Pain ^i^	30-day Postoperative Complication ^i^	Bowel Resection ^i^	Reoperation for Recurrence ^ii^
Observations (n)	**5023**	**2379**	**5023**	**5023**	**5023**
Age median (years)					
15–65	1 (ref)	1	1	1	1
65+	**5.07 (1.83, 14.09)**	**0.78 (0.62, 0.98)**	1.15 (0.97, 1.36)	**1.76 (1.26, 2.46)**	0.9 (0.61, 1.32)
Gender					
Men	1	1	1	1	1
Women	0.72 (0.42, 1.23)	1.01 (0.74, 1.37)	0.83 (0.67, 1.02)	1.22 (0.91, 1.65)	1.08 (0.64, 1.83)
ASA fitness grade					
I–II	1	1	1	1	1
III–IV	**5.75 (3.5, 9.42)**	**1.22 (0.98, 1.52)**	**1.48 (1.28, 1.71)**	**1.6 (1.26, 2.04)**	0.75 (0.5, 1.13)
Hernia anatomy					
Direct	1	1	1	1	1
Indirect	1.06 (0.67, 1.68)	1.07 (0.84, 1.36)	1.06 (0.89, 1.25)	**0.7 (0.51, 0.96)**	0.86 (0.58, 1.28)
Femoral	1.48 (0.78, 2.81)	1.1 (0.75, 1.62)	**1.61 (1.25, 2.09)**	**2.19 (1.52, 3.15)**	0.9 (0.47, 1.71)
Combined	1.55 (0.78, 3.1)	1.04 (0.66, 1.62)	1.05 (0.79, 1.41)	0.57 (0.29, 1.13)	0.62 (0.26, 1.46)
Hernia defect size (cm)					
<1.5	1	1	1	1	1
1.5–3	1.01 (0.6, 1.7)	0.9 (0.69, 1.16)	1.15 (0.95, 1.39)	1.22 (0.92, 1.61)	1.2 (0.75, 1.9)
>3	1.26 (0.71, 2.25)	0.92 (0.68, 1.24)	**1.54 (1.25, 1.91)**	1.04 (0.72, 1.5)	1.44 (0.85, 2.41)
Surgical method of repair					
Open anterior (a) mesh	1	1	1	1	1
Laparo-endoscopic mesh	0.56 (0.17, 1.87)	0.88 (0.6, 1.29)	0.77 (0.57, 1.05)	0.53 (0.28, 1.01)	0.94 (0.47, 1.89)
Open posterior(p) mesh	1.15 (0.65, 2.01)	0.99 (0.70, 1.41)	0.95 (0.76, 1.2)	**2.69 (1.96, 3.71)**	1.12 (0.64, 1.95)
Combined a/p mesh	1.14 (0.41, 3.2)	1.24 (0.64, 2.41)	0.95 (0.62, 1.48)	**1.97 (1.12, 3.44)**	1.96 (0.8, 4.8)
Non-mesh suture	**2.15 (1.2, 3.84)**	1.24 (0.77, 1.99)	**1.38 (1.05, 1.84)**	**4.86 (3.41, 6.92)**	1.75 (0.9, 3.38)
Primary hernia repair					
Yes	1	1	1	1	1
No (recurrent repair)	0.85 (0.5, 1.44)	1.05 (0.78, 1.42)	1.10 (0.9, 1.34)	1.11 (0.81, 1.53)	**2.08 (1.36, 3.17)**
30-day postoperative complication					
No	1	1		1	1
Yes	**5.63 (3.86, 8.22)**	1.19 (0.93, 1.52)		**1.76 (1.38, 2.26)**	**2.12 (1.47, 3.05)**
Bowel resection					
No	1	1	1		1
Yes	**1.88 (1.15, 3.07)**	0.64 (0.39, 1.05)	**1.76 (1.38, 2.24)**		0.49 (0.19, 1.21)

Legend Each of the columns represents a separate analysis with a separate outcome. ^i^ Logistic regression model, OR (odds ratio) for column 1–4 with a binary outcome. ^ii^ Cox proportional hazard regression model, HR (hazard ratio) for column 5 with a binary outcome. Significant estimates are in bold.

## Data Availability

All data are uncoded and can be provided on request.

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
