# Peer review of "Short- and Long-Term Outcomes After Emergency Groin Hernia Surgery: A Nationwide Population-Based Study from the Swedish Hernia Register"

_jcm, 2025, doi:10.3390/jcm14072397_

Round 1

Reviewer 1 Report

Comments and Suggestions for Authors

 The manuscript “Short and long-term outcomes after emergency groin hernia surgery: A nationwide populations-based study from the Swe-dish Hernia Register” is a large population-based study that is well-written with clear aims.  I have reviewed papers previously from the SHR and I applaud the efforts to continue this important database going to provide good insights into important questions related to hernia surgery outcomes.

On table 1, it would be good to see “p” values.  In table 2, it you have confidence intervals, you do not need “p” values, but that is OK. 

The stamen where a population-based study is representative of an unselected group of hernias undertaken by general surgeons reduces bias compared to randomized controlled trials might be true.  However, RCTs and population-based studies often attempt to answer different questions, and each has a validity that should be viewed as complementary.  I would revise this statement.   

I read the conclusions last, and the authors suggest similar issues as the observations of this reviewer, please consider these statements in the body of the manuscript and/or the discussion. 

It is important to note that the watchful-waiting approach for inguinal hernias applies ONLY to inguinal hernias in MEN.  Female patients or any patient with a femoral hernia should be offered an operation as soon as diagnosis is made.  Older patients with high suspicion of femoral hernias should undergo imaging to exclude a femoral hernia.  If a femoral hernia is present, it should be repaired electively.  If an inguinal hernia is present in a non-symptomatic elderly patient is found, the discussion about the risks and benefits of an operation considering these results could be presented to the patient. 

Author Response

Comments 1: The manuscript “Short and long-term outcomes after emergency groin hernia surgery: A nationwide populations-based study from the Swe-dish Hernia Register” is a large population-based study that is well-written with clear aims.  I have reviewed papers previously from the SHR and I applaud the efforts to continue this important database going to provide good insights into important questions related to hernia surgery outcomes. 

Anver 1: 

Thank you very much for this kind comment.

Comment 2: On table 1, it would be good to see “p” values.  In table 2, it you have confidence intervals, you do not need “p” values, but that is OK. 

Answer 2: 

Thank you for this comment. P-values are now added in Table 1.

Comment 3: The statement where a population-based study is representative of an unselected group of hernias undertaken by general surgeons reduces bias compared to randomized controlled trials might be true.  However, RCTs and population-based studies often attempt to answer different questions, and each has a validity that should be viewed as complementary.  I would revise this statement. 

Answer to comment 3: Thank you for this comment. We are in agreement with you in that RCTs and large population-based studies are both valuable and to be considered complementary. We have revised the statement and refrain from comparison to RCTs.

Comment 4: 

I read the conclusions last, and the authors suggest similar issues as the observations of this reviewer, please consider these statements in the body of the manuscript and/or the discussion. 

It is important to note that the watchful-waiting approach for inguinal hernias applies ONLY to inguinal hernias in MEN.  Female patients or any patient with a femoral hernia should be offered an operation as soon as diagnosis is made.  Older patients with high suspicion of femoral hernias should undergo imaging to exclude a femoral hernia.  If a femoral hernia is present, it should be repaired electively.  If an inguinal hernia is present in a non-symptomatic elderly patient is found, the discussion about the risks and benefits of an operation considering these results could be presented to the patient. 

Answer to comment 4: 

Thank you for these suggestions. We have added that the watchful-waiting studies apply only to men and that femoral hernias should be repaired. This has been revised and highlighted in yellow in the manuscript.

Reviewer 2 Report

Comments and Suggestions for Authors

A Review report on Short and long-term outcomes after emergency groin hernia surgery: A nationwide populations-based study from the Swedish Hernia Register

  • A brief summary: Authors checked prospectively maintained Swedish Hernia Register to evaluate and compare 30-day mortality, patient-reported chronic pain, 30-day postoperative complication, bowel resection during the emergency hernia repair and reoperation for recurrence between elective and emergency groin hernia repairs. The strengths are the number of the patients and prospectively maintained database.
  • General concept comments
    Article: the article is interesting, well-written, but is needs improvements and clarifications.

Specific comments:

  • Abstract: Key risk-factors for the outcomes – what outcomes?

Conclusion (abstract): Emergency hernia surgery continues to pose a high risk of mortality and morbidity in current practice and should be particularly avoided in elderly, comorbid patients and those with potential femoral hernias – this must be rewritten. An incarcerated hernia is an emergency which is fatal when not resolved.

Introduction:

  • Introduction is well-written, but:
    • … whereas presence of bowel resection can be a surgical measurement of the [stangulation]. – it is not true. If the presentation is within 6 hours, bowel resection is not necessary.
    • The aim must be rewritten to point out the outcomes of the emergency repairs.
  • Materials and methods:
    • Complications should be reported in a Clavien-Dindo grading system which enables comparing of outcomes: 10.1097/01.sla.0000133083.54934.ae.
  • Results:
    • Page 6: Description of ASA belongs to Materials and Methods;
    • also: iv Individuals reporting an Inguinal Pain Questionnaire (IPQ) item 2 score greater than 3 are classified as experiencing chronic postoperative pain. ivYes+no = responders, non-responders är missing values. belongs to Materials and Methods.
    • Table 2: the title of the table is weird; I cannot understand what all that mean.
  • Almost all bowel resections in the cohort were performed within the emergency repair population (98.2%), with a total rate of 7.8% of the emergency repairs compared to 0% for elective repairs (Table 1). – it must be checked and rewritten.
  • The likelihood of requiring bowel resection during emergency repairs was associated with age over 65 years, ASA fitness grade III-V, femoral hernias, open anterior repairs, combined mesh repair, suture repair and the presence of 30-day postoperative complication. – it is also nonsense mixing up causes and consequences.
  • Table 3 is a incomprehensible mess which must be separated into two tables; one for logistic regression analysis and one for Cox proportional hazard regression model. Think carefully which outcome really needs this analysis. Furthermore, logistic regression analysis is incomplete without ROC analysis.
  • Discussion:
    • Generally, this boasting of SHR being the largest, the most complete and so on is unnecessary and gives the Discussion a cheap impression.
    • ... showing that patients undergoing elective hernia repair experience a lower mortality rate compared to the background population.What is a background population?
    • .  The results indicated a higher prevalence of chronic pain in patients who had emergency surgery compared to those who had elective procedures. However, the study’s results are in contrast with those of Dahlstrand et al., who found that femoral emergency hernia repairs were associated with a decreased risk of pain. It is important to note that Dahlstrand et al. focused exclusively on femoral hernias. In this large-scale cohort, patients with femoral emergency hernias did not report less pain than those with inguinal hernias. Furthermore, there are very few studies, if any, that present data on patient-reported outcomes following emergency hernia surgery. – What did you want to say at all? Anyway, the rates should be reported.
    • … Among the emergency cohort, laparo-endoscopic techniques were associated with a lower rate of bowel resection, whereas non-mesh suture repairs were linked to increased mortality and bowel resection rates. - This paragraph must be rewritten as it gives impression that surgical approach dictate the bowel resection and not the fact of bowel strangulation and the duration of it.
    • Previous studies have consistently shown that emergency hernia repairs are associated with poorer outcomes, suggesting that they should be avoided when possible.How can we avoid surgery for irreducible, incarcerated hernia?
    • However, the question of whether elective hernia repair can effectively reduce the risk of emergency situations remains debatable.It is doubtful. If hernia is repaired, it cannot get incarcerated.
    • The significant number of unselected groin hernia repairs conducted by general surgeons with varying levels of expertise reduces the potential single-surgeon bias, thereby enhancing the external validity of the findings, presenting a real-life setting in contrast to randomized controlled trials whereas repairs are usually performed by experts. – It is an unnecessary and nonsense paragraph. What is external validity, if you have one cohort? You also cannot conduct randomized controlled trials on emergency surgical settings.
  • Conclusion must be rewritten, because it summarizes all aforementioned weird statements.

General questions to help guide your review report for research articles:

  • Is the manuscript clear – it must be improved; relevant for the field – yes, and presented in a well-structured manner? – it must be improved.
  • Are the cited references mostly recent publications (within the last 5 years) and relevant? Yes. Does it include an excessive number of self-citations? No.
  • Is the manuscript scientifically sound and is the experimental design appropriate to test the hypothesis? Yes.
  • Are the manuscript’s results reproducible based on the details given in the methods section? Yes.
  • Are the figures/tables/images/schemes appropriate? No. Do they properly show the data? No. Are they easy to interpret and understand? No. Is the data interpreted appropriately and consistently throughout the manuscript? No. Please include details regarding the statistical analysis or data acquired from specific databases. Table 3 must be solved out.
  • Are the conclusions consistent with the evidence and arguments presented? No.
  • Please evaluate the ethics statements and data availability statements to ensure they are adequate. They are ok.

Author Response

Comment 1:  Abstract: Key risk-factors for the outcomes – what outcomes?

Response 1:  Thank you for this comment. We refer to the outcomes listed in the previous sentence. However, as the sentence can leave room for interpretation, we have rephrased it and it has been highlighted in yellow.

Comment 2:  Conclusion (abstract): Emergency hernia surgery continues to pose a high risk of mortality and morbidity in current practice and should be particularly avoided in elderly, comorbid patients and those with potential femoral hernias – this must be rewritten. An incarcerated hernia is an emergency which is fatal when not resolved.

Response 2: Thank you for noticing that the wording was unclear and could be misinterpreted. We do not mean that one should refrain from operating once the emergency has arisen, but rather that we need to do what we can to minimize the number of patients who experience acute incarceration – especially in the frail patients and those with femoral hernias. The conclusion has been changed to: Emergency hernia surgery continues to pose a high risk of mortality and morbidity. Elective repair should be considered in frail patients and those with potential femoral hernias.

Comment 3: Introduction is well-written, but: … whereas presence of bowel resection can be a surgical measurement of the [stangulation]. – it is not true. If the presentation is within 6 hours, bowel resection is not necessary.

Response 3: Thank you for bringing the over-simplification to our attention. We have revised the text to: “While bowel resection can often be avoided if the strangulation has lasted shorter than six hours, the need for resection can be used as an indication of irreversible strangulation ischemia.”

Comment 4The aim must be rewritten to point out the outcomes of the emergency repairs.

Response 4: We have rephrased the aim to: This study aimed to evaluate and compare emergency and elective groin hernia repairs regarding: 30-day mortality; patient-reported chronic pain 1-year post-surgery; 30-day postoperative complication; bowel resection; and reoperation for recurrence.

Comment 5: Materials and methods: Complications should be reported in a Clavien-Dindo grading system which enables comparing of outcomes: 10.1097/01.sla.0000133083.54934.ae.

Response 5:  We agree that reporting complication severity according to Clavien-Dindo would have been desirable. Unfortunately, recording according to this system was first introduced in the SHR in August 2015, thus that information was not available for operations performed during the first half of the study period.

Comment 6: Page 6: Description of ASA belongs to Materials and Methods;

Response 6: We have added the description of ASA to the materials and methods section regarding the study population and registered variables. However, we have also left it where it was (as part of the legend to Table 1) in order for the reader to be able to interpret the Table independently of text. It can be removed from there at the editor’s discretion.

Comment 7: also: iv Individuals reporting an Inguinal Pain Questionnaire (IPQ) item 2 score greater than 3 are classified as experiencing chronic postoperative pain. ivYes+no = responders, non-responders är missing values. belongs to Materials and Methods.

Response 7: This is also part of the legend to Table 1. It has previously been described in the section Material and methods, PROM pain questionnaire. We believe that including the information in the legend facilitates understanding for readers viewing the table, but if the editor finds it unnecessary, it can be removed from the legend.

Comment 8: Table 2: the title of the table is weird; I cannot understand what all that mean.

Response 8: Thank you for bringing to our attention that the table title was difficult to interpret. We have changed the title to “Risk of 30-day mortality, chronic pain, 30-day postoperative complication, bowel resection and reoperation for recurrence – multivariable regression analyses”.

Comment 9: Almost all bowel resections in the cohort were performed within the emergency repair population (98.2%), with a total rate of 7.8% of the emergency repairs compared to 0% for elective repairs (Table 1). – it must be checked and rewritten.

Response 9: Thank you for advising us that this can be unclear to the reader. While 1.8% of all resections were made in elective procedures, that equals 7 out of 89,212 elective repairs (78 parts per million). In order to clarify the sentence, we have deleted the part reporting the percentage of emergency procedures among those who had a bowel resection. The sentence now reads: “Almost all bowel resections in the cohort were performed within the emergency repair population, with a total rate of 7.8% of the emergency repairs compared to 0% for elective repairs (Table 1).”

Comment 10: The likelihood of requiring bowel resection during emergency repairs was associated with age over 65 years, ASA fitness grade III-V, femoral hernias, open anterior repairs, combined mesh repair, suture repair and the presence of 30-day postoperative complication. – it is also nonsense mixing up causes and consequences.

Response 10:  In the discussion we discuss the drawbacks of observational studies. In observational studies it is not possible to determine causes or effects. Hence, we have not defined any of the found associations as causes or consequences. Whatever interpretation of the associations, has to be left to the discussion. In the first paragraph of the discussion, the only variables we mention there, as having an increased risk for bowel resection, are those that cannot be a consequence of the bowel resection (ie age, ASA and hernia anatomy rather than repair method or postoperative complications).

Comment 11: Table 3 is an incomprehensible mess which must be separated into two tables; one for logistic regression analysis and one for Cox proportional hazard regression model. Think carefully which outcome really needs this analysis. Furthermore, logistic regression analysis is incomplete without ROC analysis.

Response 11:  Thank you for your feedback regarding Table 3. To enhance clarity for the reader, we have adjusted the column widths so that the confidence intervals are presented on a single line. Additionally, we have indicated in the legend that each of the columns represents a separate analysis.

In striving for brevity and avoiding redundancy, we decided to present all analyses in one table, even though they include different regression methods. Our primary aim with Table 3 was to elucidate the relationships between risk factors and the different outcomes, and we believe that logistic regression provides meaningful insights, particularly with the odds ratios and hazard ratios associated with each risk factor.

Given that this is a national cohort, we feel confident that we have a robust representation of the population. While ROC analysis is indeed a valuable tool for evaluating predictive performance, however our focus here is primarily on understanding the relationships between the variables and the outcomes. In this setting we do not believe that including ROC curves will add greatly to the understanding for the reader. However, the analyses are present and the AUCs (area under the curve) for the ROC curves for respective analyses in table 2 were as follows: 30-day mortality, AUC= 0.9538; chronic pain, AUC=0.5601; postoperative complication, AUC=0.6097; bowel resection, AUC=0.9851 and for the analyses in table 3 they were: 30-day mortality, AUC=0.8637; chronic pain, AUC=0.5486; postoperative complication, AUC=0.6054; bowel resection, AUC=0.7875. If the editors despite our argument above find that the ROC analysis would be of interest, we can provide them as supplementary material. HH (one of the authors) previously served as the representative biostatistician for the Swedish Hernia Register and have performed all the statistical analyses.

Comment 12: Discussion: Generally, this boasting of SHR being the largest, the most complete and so on is unnecessary and gives the Discussion a cheap impression.

Response 12: We have taken part of your opinion. We certainly did not mean to offend by our description of the study or the register itself. While we feel that we have tried to present the work in a fairly balanced manner, we do respect that you are of a different opinion and that there may be others that would feel the same way. Upon several careful read-throughs we have identified that there were two paragraphs regarding postoperative pain where the large size of the cohort was mentioned. We have rephrased the first sentences in the paragraph about chronic pain on page 10 to avoid that reiteration. The other paragraph where it is mentioned is in the paragraph describing strengths of the study, where a discussion of sample size might be considered natural. We hope that this will be found more palatable.

Comment 13:  ...showing that patients undergoing elective hernia repair experience a lower mortality rate compared to the background population. – What is a background population?

Response 13: Thank you for highlighting this unclarity. This phrase refers to the article “Mortality after groin hernia surgery”. In this article the outcome is standardized mortality ratio (SMR), where data on mortality of men and women at different ages were collected from Statistics of Sweden. The mortality seen in the study cohort was then related to the expected mortality for the age- and sex-matched “background” population. We have revised the sentence to reflect that.

Comment 14: ….  The results indicated a higher prevalence of chronic pain in patients who had emergency surgery compared to those who had elective procedures. However, the study’s results are in contrast with those of Dahlstrand et al., who found that femoral emergency hernia repairs were associated with a decreased risk of pain. It is important to note that Dahlstrand et al. focused exclusively on femoral hernias. In this large-scale cohort, patients with femoral emergency hernias did not report less pain than those with inguinal hernias. Furthermore, there are very few studies, if any, that present data on patient-reported outcomes following emergency hernia surgery. – What did you want to say at all? Anyway, the rates should be reported.

Response 14:  Long-term pain after emergency groin hernia repair is not well studied, so there are not many earlier studies to reflect the results against. There is one publication from 2011 (Dahlstrand et al, reference 21) where 1461 patients who had a femoral hernia repair responded to questions about pain in the operated groin. In that study patients who had emergency repairs reported less pain than those who had elective repairs. We believe it is useful and important to address the discrepancies on the matter against other studies. However, the paragraph has been slightly revised and highlighted in yellow in the manuscript. The rates are reported in the results section.

Comment 15: … Among the emergency cohort, laparo-endoscopic techniques were associated with a lower rate of bowel resection, whereas non-mesh suture repairs were linked to increased mortality and bowel resection rates. - This paragraph must be rewritten as it gives impression that surgical approach dictate the bowel resection and not the fact of bowel strangulation and the duration of it.

Response 15: We agree that it is highly likely that the surgical approach was dictated by the need for bowel resection, rather than the other way around. The following sentence was supposed to convey just that, but we agree that it could be more clearly stated. We have rephrased that (following) sentence to: “This disparity is likely to be attributed to selection of surgical method according to severity of the case, where less complicated and less contaminated emergency hernias were treated with laparo-endoscopic techniques, while non-mesh suture repairs were favored in more complex and contaminated cases”.

Comment 16: Previous studies have consistently shown that emergency hernia repairs are associated with poorer outcomes, suggesting that they should be avoided when possible. – How can we avoid surgery for irreducible, incarcerated hernia?

Response 16: Thank you for making it clear that our phrasing can be misinterpreted. We intended to shine a light on the fact that elective repair should be considered in patients where the risk for acute surgery can be high, as well as in patients that can be considered to be healthy enough to undergo elective surgery but frail enough to not be able to tolerate a hernia emergency. We have rephrased the sentence, aiming to clarify that: “Previous studies have consistently shown that emergency hernia repairs are associated with poorer outcomes, suggesting that we should actively work to lower the number of hernia emergencies”.

Comment 17: However, the question of whether elective hernia repair can effectively reduce the risk of emergency situations remains debatable. – It is doubtful. If hernia is repaired, it cannot get incarcerated.

Response 17: While that may be true, it still remains open for debate to which extent a more “aggressive” approach repairing all diagnosed groin hernias, irrespective of symptoms, would lower the rate of emergencies. In addition to the large randomized trials on minimally symptomatic hernias in males that demonstrated a very low number of emergencies in the watchful waiting groups, it has also been found that many of the patients presenting with a femoral hernia emergency had not previously been diagnosed with the hernia and a rather large group had not had any symptoms from that groin prior to the emergency. That being said, as stated under previous comments, we believe that surgeons should carefully consider the risk of future acute incarceration and the individual patient’s capability to cope with such a situation (frailty) when deciding on elective repair vs expectancy.

Comment 18: The significant number of unselected groin hernia repairs conducted by general surgeons with varying levels of expertise reduces the potential single-surgeon bias, thereby enhancing the external validity of the findings, presenting a real-life setting in contrast to randomized controlled trials whereas repairs are usually performed by experts. – It is an unnecessary and nonsense paragraph. What is external validity, if you have one cohort? You also cannot conduct randomized controlled trials on emergency surgical settings.

Response 18: We are not quite sure why you find this part of the discussion unnecessary. The point is describing a strength of the study (limitations are addressed in another paragraph) and is included to aid the reader in their interpretation of our findings – when they themselves can weigh strengths against limitations. More specifically, external validity addresses whether the findings of a study can be applied in other contexts. Population validity is one type of external validity, ie whether the findings can be generalized to other groups of people. The fact that we have not excluded age-groups, limited us to only certain procedures or only included specific type of hospitals etc, but rather included the entire population, would suggest that our results are more possible to extrapolate to at least countries that are somewhat similar to Sweden’s access to health care.

Emergency procedures can be difficult to study in randomized controlled trials, which is one of the reasons that we think a study such as this can be of value. RCTs and population-based observational studies can be considered to complement each other in describing realities in a scientific manner. We have revised the sentence.

Comment 19: Conclusion must be rewritten, because it summarizes all aforementioned weird statements. 

Response 19:  The conclusion has been revised to avoid the misunderstanding that we are advocating to not perform emergency surgery once the emergency has arisen.

Reviewer 3 Report

Comments and Suggestions for Authors

The paper, "Short and long-term outcomes after emergency groin hernia surgery: A nationwide populations-based study from the Swedish Hernia Register," is a wonderful paper that uses the Swedish Hernia Register to look at emergent vs elective hernia repair for common short and long-term complications.

Overall, there are few faults with this paper.  The introduction is thorough yet succinct. The study design is sound, including well-thought-out inclusion and exclusion criteria. The sample size is large and diverse. The author's analysis of the results is insightful and informative.

My only recommendations are grammatical/formative.

  1.  The subheading "Statistical analysis" is not formatted the same as other subheadings. 
  2. Under the subheading "Reopration for recurrence," it reads, "...with a non-significantly, increased HR at 1.14 (95% CI 0.95-1.38) for emergency repairs (Table 2)."  I assume this is meant to be OR (as all previous paragraphs discuss odd ratios). 

Otherwise, excellent work. 

Author Response

Comment 1: The paper, "Short and long-term outcomes after emergency groin hernia surgery: A nationwide populations-based study from the Swedish Hernia Register," is a wonderful paper that uses the Swedish Hernia Register to look at emergent vs elective hernia repair for common short and long-term complications.

Overall, there are few faults with this paper.  The introduction is thorough yet succinct. The study design is sound, including well-thought-out inclusion and exclusion criteria. The sample size is large and diverse. The author's analysis of the results is insightful and informative.

Response 1: Thank you for these kind words.

Comment 2:  My only recommendations are grammatical/formative.

  1.  The subheading "Statistical analysis" is not formatted the same as other subheadings. 
  2. Under the subheading "Reopration for recurrence," it reads, "...with a non-significantly, increased HR at 1.14 (95% CI 0.95-1.38) for emergency repairs (Table 2)."  I assume this is meant to be OR (as all previous paragraphs discuss odd ratios). 

Otherwise, excellent work. 

Response 2: Thank you for noticing the formatting error about “statistical analysis”. It has now been corrected.

When it comes to the reporting on reoperation for recurrence, we have chosen to report this outcome using a time-to-event analysis (Cox regression) rather than a logistic regression model. Since the follow-up period varies between the study objects - a method which takes this into account, whether a recurrence occurs after 1, 4 or 7 years for example, was considered to be more informative and statistical correct than to report on recurrence at a specific time (say 5 years post-surgery or so) with a logistic regression model with odds ratios. To aid the reader we have now written out HR as ‘hazard ratio’ in that paragraph.

In summary, thank you again for the reviewers’ questions and comments. We sincerely hope that we have adequately addressed your comments and made the necessary improvements align with the referee´s important comments.

Kind regards,

All the authors

Round 2

Reviewer 2 Report

Comments and Suggestions for Authors

The authors significantly improved the manuscript and answered on raised questions.